# Identifying Outlier Arms in Multi-Armed Bandit *

**Honglei Zhuang**[1][†]      **Chi Wang**[2]      **Yifan Wang**[3]
[1]University of Illinois at Urbana-Champaign
[2]Microsoft Research, Redmond
[3]Tsinghua University
hzhuang3@illinois.edu      wang.chi@microsoft.com
yifan-wa16@mails.tsinghua.edu.cn

## Abstract

We study a novel problem lying at the intersection of two areas: multi-armed bandit and outlier detection. Multi-armed bandit is a useful tool to model the process of incrementally collecting data for multiple objects in a decision space. Outlier detection is a powerful method to narrow down the attention to a few objects after the data for them are collected. However, no one has studied how to *detect outlier objects while incrementally collecting data for them*, which is necessary when data collection is expensive. We formalize this problem as identifying outlier arms in a multi-armed bandit. We propose two sampling strategies with theoretical guarantee, and analyze their sampling efficiency. Our experimental results on both synthetic and real data show that our solution saves 70-99% of data collection cost from baseline while having nearly perfect accuracy.

## 1   Introduction

A multi-armed bandit models a set of items (arms), each associated with an unknown probability distribution of rewards. An observer can iteratively select an item and request a sample reward from its distribution. This model has been predominant in modeling a broad range of applications, such as cold-start recommendation [24], crowdsourcing [13] *etc.* In some applications, the objective is to maximize the collected rewards while playing the bandit (exploration-exploitation setting [7, 5, 23]); in others, the goal is to identify an optimal object among multiple candidates (pure exploration setting [6]).

In the pure exploration setting, rich literature is devoted to the problem of identifying the top-$K$ arms with largest reward expectations [8, 15, 20]. We consider a different scenario, in which one is more concerned about "outlier arms" with extremely high/low expectation of rewards that substantially deviate from others. Such arms are valuable as they usually provide novel insight or imply potential errors.

For example, suppose medical researchers are testing the effectiveness of a biomarker X (*e.g.*, the existence of a certain gene sequence) in distinguishing several different diseases with similar

†Part of this work was done while the first author was an intern at Microsoft Research. The first author was sponsored in part by the U.S. Army Research Lab. under Cooperative Agreement No. W911NF-09-2-0053 (NSCTA), National Science Foundation IIS 16-18481, IIS 17-04532, and IIS-17-41317, and grant 1U54GM114838 awarded by NIGMS through funds provided by the trans-NIH Big Data to Knowledge (BD2K) initiative (www.bd2k.nih.gov). The views and conclusions contained in this document are those of the author(s) and should not be interpreted as representing the official policies of the U.S. Army Research Laboratory or the U.S. Government. The U.S. Government is authorized to reproduce and distribute reprints for Government purposes notwithstanding any copyright notation hereon.

symptoms. They need to perform medical tests (*e.g.*, gene sequencing) on patients with each disease of interest, and observe if X's degree of presence is significantly higher in a certain disease than other diseases. In this example, a disease can be modeled as an arm. The researchers can iteratively select a disease with which they sample a patient and perform the medical test to observe the presence of X. The reward is 1 if X is fully present, and 0 if fully absent. To make sure the biomarker is useful, researchers look for the disease with an extremely high expectation of reward compared to other diseases, instead of merely searching for the disease with the highest reward expectation. The identification of "outlier" diseases is required to be sufficiently accurate (*e.g.*, correct with 99% probability). Meanwhile, it should be achieved with a minimal number of medical tests in order to save the cost. Hence, a good sampling strategy needs to be developed to both guarantee the correctness and save cost.

As a generalization of the above example, we study a novel problem of identifying outlier arms in multi-armed bandits. We define the criterion of outlierness by extending an established rule of thumb, *3σ rule*. The detection of such outliers requires calculating an outlier threshold that depends on the mean reward of all arms, and outputing outlier arms with an expected reward above the threshold. We specifically study pure exploration strategies in a fixed confidence setting, which aims to output the correct results with probability no less than $1 - \delta$.

Existing methods for top-$K$ arm identification cannot be directly applied, mainly because the number of outliers are unknown *a priori*. The problem also differs from the thresholding bandit problem [26], as the outlier threshold depends on the (unknown) reward configuration of all the arms, and hence also needs to be explored. Given the outlierness criterion, the key challenges in tackling this problem are: i) how to guarantee the identified outlier arms truly satisfy the criterion; and ii) how to design an efficient sampling strategy which balances the trade-off between exploring individual arms and exploring outlier threshold.

In this paper, we make the following major contributions:

- We propose a Round-Robin sampling algorithm, with a theoretical guarantee of its correctness as well as a theoretical upper bound of its total number of pulls.
- We further propose an improved algorithm Weighted Round-Robin, with the same correctness guarantee, and a better upper bound of its total number of pulls.
- We verify our algorithms on both synthetic and real datasets. Our Round-Robin algorithm has near 100% accuracy, while reducing the cost of a competitive baseline up to 99%. Our Weighted Round-Robin algorithm further reduces the cost by around 60%, with even smaller error.

## 2    Related Work

We present studies related to our problem in different areas.

**Multi-armed bandit.**    Multi-armed bandit is an extensively studied topic. A classic setting is to regard the feedback of pulling an arm as a reward and aim to optimize the exploration-exploitation trade-off [7, 5, 23]. In an alternative setting, the goal is to identify an optimal object using a small cost, and the cost is related to the number of pulls rather than the feedback. This is the "pure exploration" setting [6]. Early work dates back to 1950s under the subject of sequential design of experiments [27]. Recent applications in crowdsourcing and big data-driven experimentation *etc.* revitalized this field. The problem we study also falls into the general category of pure exploration bandit.

Within this category, a number of studies focus on best arm identification [4, 6, 14, 15], as well as finding top-$K$ arms [8, 15, 20]. These studies focus on designing algorithms with probabilistic guarantee of finding correct top-$K$ arms, and improving the number of pulls required by the algorithm. Typical cases of study include: (a) fixed confidence, in which the algorithm needs to return correct top-$K$ arms with probability above a threshold; (b) fixed budget, in which the algorithm needs to maximize the probability of correctness within a certain number of pulls. While there are promising advances in recent theoretical work, optimal algorithms in general cases remain an open problem. Finding top-$K$ arms is different from finding outlier arms, because top arms are not necessarily outliers. Yet the analysis methods are useful and inspiring to our study.

There are also studies [26, 11] on thresholding bandit problem, where the aim is to find the set of arms whose expected rewards are larger than a given threshold. However, since the outlier threshold

depends on the unknown expected rewards of all the arms, these algorithms cannot apply to our problem.

Some studies [12, 16] propose a generalized objective to find the set of arms with the largest sum of reward expectations with a given combinatorial constraint. The constraint is independent of the rewards (*e.g.*, the set must have $K$ elements). Our problem is different as the outlier constraint depends on the reward configuration of all the arms.

A few studies on clustering bandits [17, 22] aim to identify the internal cluster structure between arms. Their objective is different from outlier detection. Moreover, they do not study a pure-exploration scenario.

Carpentier and Valko [9] propose the notion of "extreme bandits" to detect a different kind of outlier: They look for extreme values of individual rewards from each pull. Using the medical example in Section 1, the goal can be interpreted as finding a patient with extremely high containment of a biomarker. With that goal, the arm with the heaviest tail in its distribution is favored, because it is more likely to generate extremely large rewards than other arms. In contrast, our objective is to find arms with extremely large expectations of rewards.

**Outlier detection.** Outlier detection has been studied for decades [10, 18]. Most existing work focuses on finding outlier data points from *observed* data points in a dataset. We do not target on finding outlier data points from observed data points (rewards). Instead, we look for outlier arms which generate these rewards. Also, these rewards are not provided at the beginning to the algorithm, and the algorithm needs to proactively pull each arm to obtain more reward samples.

Sampling techniques were used in detecting outlier data points from observed data points with very different purposes. In [1], outlier detection is reduced to a classification problem and an active learning algorithm is proposed to selectively sample data points for training the outlier detector. In [28, 29], a subset of data points is uniformly sampled to accelerate the outlier detector. Kollios et al. [21] propose a biased sampling strategy. Zimek et al. [30], Liu et al. [25] use subsampling technique to introduce diversity in order to apply ensemble methods for better outlier detection performance. In outlier arm identification, the purpose of sampling is to estimate the reward expectation of each arm, which is a hidden variable and can only be estimated from sampled rewards.

There are also studies on outlier detection when uncertainty of data points is considered [2, 19]. However, these algorithms do not attempt to actively request more information about data points to reduce the uncertainty, which is a different setting from our work.

## 3 Problem Definition

In this section, we describe the problem of identifying outlier arms in a multi-armed bandit. We start with recalling the settings of the multi-armed bandit model.

**Multi-armed bandit.** A multi-armed bandit (MAB) consists of $n$-arms, where each arm is associated with a reward distribution. The (unknown) expectation of each reward distribution is denoted as $y_i$. At each iteration, the algorithm is allowed to select an arm $i$ to play (pull), and obtain a sample reward $x_i^{(j)} \in \mathcal{R}$ from the corresponding distribution, where $j$ corresponds to the $j$-th samples obtained from the $i$-th arm. We further use $\mathbf{x}_i$ to represent all the samples obtained from the $i$-th arm, and $\mathbf{y}$ to represent the configuration of all the $y_i$'s.

**Problem definition.** We study to identify outlier arms with extremely high reward expectations compared to other arms in the bandit. To define "outlier arms", we adopt a general statistical rule named $k$-sigma: The arms with reward expectations higher than the mean plus $k$ standard deviation of all arms are considered as outliers. Formally, we define the mean of all the $n$ arms' reward expectations as well as their standard deviation as:

$$\mu_y = \frac{1}{n}\sum_{i=1}^{n} y_i, \quad \sigma_y = \sqrt{\frac{1}{n}\sum_{i=1}^{n}(y_i - \mu_y)^2}$$

We define a threshold function based on the above estimators as:

$$\theta = \mu_y + k\sigma_y$$

An arm $i$ is defined as an outlier arm iff $y_i > \theta$ and is defined as a normal (non-outlier) arm iff $y_i < \theta$. We denote the set of outlier arms as $\Omega = \{i \in [n] | y_i > \theta\}$.

In a multi-armed bandit setting, the value of $y_i$ for each arm is unknown. Instead, the system needs to pull one arm at each iteration to obtain a sample, and estimate the value $y_i$ for each arm and the threshold $\theta$ from all the obtained samples $\mathbf{x}_i, \forall i$. We introduce the following estimators:

$$\hat{y}_i = \frac{1}{m_i} \sum_j x_i^{(j)}, \quad \hat{\mu}_y = \frac{1}{n} \sum_{i=1}^{n} \hat{y}_i, \quad \hat{\sigma}_y = \sqrt{\frac{1}{n} \sum_{i=1}^{n} (\hat{y}_i - \hat{\mu}_y)^2}, \quad \hat{\theta} = \hat{\mu}_y + k\hat{\sigma}_y$$

where $m_i$ is the number of times the arm $i$ is pulled.

We focus on the *fixed confidence* setting. The objective is to design an efficient pulling algorithm, such that the algorithm can return the true set of outlier arms $\Omega$ with probability at least $1 - \delta$ ($\delta$ is usually a small constant). The fewer total number of pulls, *i.e.* $T = \sum_i m_i$, the better, because each pull has a economic or time cost. Note that this is a *pure exploration* setting, i.e., the reward incurred during exploration is irrelevant to the cost.

# 4 Algorithms

In this section, we propose several algorithms, and present the theoretical guarantee of each algorithm.

## 4.1 Round-Robin Algorithm

The most simple algorithm is to pull arms in a round-robin way. That is, the algorithm starts from arm $1$ and pulls arm $2, 3, \cdots$ respectively, and goes back to arm $1$ after it iterates over all the $n$ arms. The process continues until a certain termination condition is met.

Intuitively, the algorithm should terminate when it is confident about whether each arm is an outlier. We achieve this by using the confidence interval of each arm's reward expectation as well as the confidence interval of the outlier threshold. If the significance levels of these intervals are carefully set, and each reward expectation's confidence interval has no overlap with the threshold's confidence interval, we can safely terminate the algorithm while guaranteeing correctness with desired high probability. In the following, we first discuss the formal definition of confidence intervals, as well as how to set the significance levels. Then we present the formal termination condition.

**Confidence intervals.** We provide a general definition of *confidence intervals* for $\hat{y}_i$ and $\hat{\theta}$. The confidence interval for $\hat{y}_i$ at significance level $\delta'$ is defined as $[\hat{y}_i - \beta_i(m_i, \delta'), \hat{y}_i + \beta_i(m_i, \delta')]$, such that:

$$\mathbb{P}(\hat{y}_i - y_i > \beta_i(m_i, \delta')) < \delta', \quad \text{and} \quad \mathbb{P}(\hat{y}_i - y_i < -\beta_i(m_i, \delta')) < \delta'$$

Similarly, the confidence interval for $\hat{\theta}$ at significance level $\delta'$ is defined as $[\hat{\theta} - \beta_\theta(\mathbf{m}, \delta'), \hat{\theta} + \beta_\theta(\mathbf{m}, \delta')]$, such that:

$$\mathbb{P}(\hat{\theta} - \theta > \beta_\theta(\mathbf{m}, \delta')) < \delta', \quad \text{and} \quad \mathbb{P}(\hat{\theta} - \theta < -\beta_\theta(\mathbf{m}, \delta')) < \delta'$$

The concrete form of confidence interval may vary with the reward distribution associated with each arm. We defer the discussion of concrete form of confidence interval to Section 4.3.

In our algorithm, we update the significance level $\delta'$ for the above confidence intervals at each iteration. After $T$ pulls, the $\delta'$ should be set as:

$$\delta' = \frac{6\delta}{\pi^2 (n+1) T^2} \tag{1}$$

In the following discussion, we omit the parameters in $\beta_i$ and $\beta_\theta$ when they are clear from the context.

**Active arms.** At any time, if $\hat{y}_i$'s confidence interval overlaps with $\hat{\theta}$'s confidence interval, then the algorithm cannot confidently tell if the arm $i$ is an outlier or a normal arm. We call such arms *active*, and vice versa. Formally, an arm $i$ is active, denoted as $\text{ACTIVE}_i = \text{TRUE}$, iff

$$\begin{cases} \hat{y}_i - \beta_i < \hat{\theta} + \beta_\theta, & \text{if } \hat{y}_i > \hat{\theta}; \\ \hat{y}_i + \beta_i > \hat{\theta} - \beta_\theta, & \text{otherwise.} \end{cases} \tag{2}$$

We denote the set of active arms as $A = \{i \in [n] | \text{ACTIVE}_i = \text{TRUE}\}$. With this definition, the termination condition is simply $A = \emptyset$. When this condition is met, we return the result set:

$$\hat{\Omega} = \{i | \hat{y}_i > \hat{\theta}\} \tag{3}$$

The algorithm is outlined in Algorithm 1.

---

**Algorithm 1:** Round-Robin Algorithm (RR)

---

**Input**: $n$ arms, outlier parameter $k$
**Output**: A set $\hat{\Omega}$ of outlier arms

1  Pull each arm $i$ once $\forall i \in [n]$;                                            // Initialization
2  $T \leftarrow n$;
3  Update $\hat{y}_i, m_i, \beta_i, \forall i \in [n]$ and $\hat{\theta}, \beta_\theta$;
4  $i \leftarrow 1$;
5  **while** $A \neq \emptyset$ **do**
6     $i \leftarrow i\%n + 1$;                                           // Round-robin
7     Pull arm $i$;
8     $T \leftarrow T + 1$;
9     Update $\hat{y}_i, m_i, \beta_i$ and $\hat{\theta}, \beta_\theta$;
10 **return** $\hat{\Omega}$ according to Eq. (3);

---

**Theoretical results.** We first show that if the algorithm terminates with no active arms, the returned outlier set will be correct with high probability.

**Theorem 1** (Correctness). *With probability $1 - \delta$, if the algorithm terminates after a certain number of pulls $T$ when there is no active arms i.e. $A = \emptyset$, then the returned set of outliers will be correct, i.e. $\hat{\Omega} = \Omega$.*

We can also provide an upper bound for the efficiency of the algorithm in a specific case when all the reward distributions are bounded within $[a, b]$ where $b - a = R$. In this case, the confidence intervals can be instantiated as discussed in Section 4.3. And we can accordingly obtain the following results:

**Theorem 2.** *With probability $1 - \delta$, the total number of pulls $T$ needed for the algorithm to terminate is bounded by*

$$T \leq 8R^2 \mathbf{H}_{RR} \left[ \log \left( \frac{2R^2 \pi^2 (n+1) \mathbf{H}_{RR}}{3\delta} \right) + 1 \right] + 4n \tag{4}$$

*where*

$$\mathbf{H}_{RR} = \mathbf{H}_1 \left( 1 + \sqrt{l(k)} \right)^2, \quad \mathbf{H}_1 = \frac{n}{\min_{i \in [n]} (y_i - \theta)^2},$$

$$l(k) = \left[ \sqrt{\frac{(1 + k\sqrt{n-1})^2}{n}} + \sqrt{\frac{k^2}{2 \log \left( \frac{\pi^2 n^3}{6\delta} \right)}} \right]^2$$

### 4.2 Weighted Round-Robin Algorithm

The round-robin algorithm evenly distributes resources to all the arms. Intuitively, active arms deserve more pulls than inactive arms, since the algorithm is almost sure about whether an inactive arm is outlier already.

Based on this idea, we propose an improved algorithm. We allow the algorithm to sample the active arms $\rho$ times as many as inactive arms, where $\rho \geq 1$ is a real constant. Since $\rho$ is not necessarily an integer, we use a method similar to stride scheduling to guarantee the ratio between number of pulls of active and inactive arms are approximately $\rho$ in a long run. The algorithm still pulls by iterating over all the arms. However, after each arm is pulled, the algorithm can decide either to stay at this arm for a few "extra pulls," or proceed to the next arm. If the arm pulled at the $T$-th iteration is the same as the arm pulled at the $(T-1)$-th iteration, we call the $T$-th pull an "extra pull." Otherwise, we call it a "regular pull." We keep a counter $c_i$ for each arm $i$. When $T > n$, after the algorithm

performs a regular pull on arm $i$, we add $\rho$ to the counter $c_i$. If this arm is still active, we keep pulling this arm until $m_i \geq c_i$ or it becomes inactive. Otherwise we proceed to the next arm to perform the next regular pull.

This algorithm is named Weighted Round-Robin, and outlined in Algorithm 2.

---

**Algorithm 2:** Weighted Round-Robin Algorithm (WRR)

**Input**: $n$ arms, outlier parameter $k$, $\rho$
**Output**: A set of outlier arms $\hat{\Omega}$

1  Pull each arm $i$ once $\forall i \in [n]$;                              `// Initialization`
2  $T \leftarrow n$;
3  Update $\hat{y}_i, m_i, \beta_i, \forall i \in [n]$ and $\hat{\theta}, \beta_\theta$;
4  $c_i \leftarrow 0, \forall i \in [n]$;
5  $i \leftarrow 1$;
6  **while** $A \neq \emptyset$ **do**
7      $i \leftarrow i\%n + 1$ ;                         `// Next regular pull`
8      $c_i \leftarrow c_i + \rho$;
9      **repeat**
10         Pull arm $i$;
11         $T \leftarrow T + 1$;
12         Update $\hat{y}_i, m_i, \beta_i$ and $\hat{\theta}, \beta_\theta$;
13     **until** $i \notin A \bigvee m_i \geq c_i$;
14 **return** $\hat{\Omega}$ according to Eq. (3);

---

**Theoretical results.** Since the Weighted Round-Robin algorithm has the same termination condition, according to Theorem 1, it has the same correctness guarantee.

We can also bound the total number of pulls needed for this algorithm when the reward distributions are bounded.

**Theorem 3.** *With probability* $1 - \delta$*, the total number of pulls* $T$ *needed for the Weighted Round-Robin algorithm to terminate is bounded by*

$$T \leq 8R^2 \mathbf{H}_{WRR}\left[\log\left(\frac{2R^2\pi^2(n+1)\mathbf{H}_{WRR}}{3\delta}\right) + 1\right] + 2(\rho + 2)n \tag{5}$$

*where*

$$\mathbf{H}_{WRR} = \left(\frac{\mathbf{H}_1}{\rho} + \frac{(\rho - 1)\mathbf{H}_2}{\rho}\right)\left(1 + \sqrt{l(k)\rho}\right)^2, \quad \mathbf{H}_2 = \sum_i \frac{1}{(y_i - \theta)^2}$$

**Determining $\rho$.** One important parameter in this algorithm is $\rho$. For bounded reward distributions, we have a closed form upper bound of $T$ as $O(\mathbf{H}_{\text{WRR}} \log \frac{\mathbf{H}_{\text{WRR}}}{\delta})$. The lower bound of $T$ is independent of $\rho$. We conjecture the lower bound to be $\Omega(\mathbf{H}_2 \log \frac{\mathbf{H}_2}{\delta})$. We aim to find the $\rho$ that minimizes the gap between the upper bound and the lower bound. We formalize the objective as finding a $\rho$ to minimize $\mathbf{H}_{\text{WRR}}/\mathbf{H}_2$. Since we do not know the reward distribution configuration $\mathbf{y}$, we use the minimax principle to find $\rho^*$ that optimizes the most difficult configuration $\mathbf{y}$, namely

$$\rho^* = \underset{\rho \geq 1}{\arg\min} \sup_{\mathbf{y}} \frac{\mathbf{H}_{\text{WRR}}}{\mathbf{H}_2}$$

Since $\frac{\mathbf{H}_1}{n} \leq \mathbf{H}_2 \leq \mathbf{H}_1$, and $\frac{\mathbf{H}_{\text{WRR}}}{\mathbf{H}_2}$ is monotonically increasing with regard to $\frac{\mathbf{H}_1}{\mathbf{H}_2}$, we can obtain the optimal value $\rho^*$ as

$$\rho^* = \frac{(n-1)^{\frac{2}{3}}}{l^{\frac{1}{3}}(k)} \tag{6}$$

**Theoretical comparison with RR.** We compare theses two algorithms by comparing their upper bounds. Essentially, we study $\mathbf{H}_{\text{WRR}}/\mathbf{H}_{\text{RR}}$ since the two bounds only differ in this term after a small

constant is ignored. We have

$$\frac{\mathbf{H}_{\text{WRR}}}{\mathbf{H}_{\text{RR}}} = \left(\frac{1}{\rho} + \frac{\rho - 1}{\rho}\frac{\mathbf{H}_2}{\mathbf{H}_1}\right)\left(\frac{1 + \sqrt{l(k)\rho}}{1 + \sqrt{l(k)}}\right)^2 \tag{7}$$

The ratio between $\mathbf{H}_2$ and $\mathbf{H}_1$ indicates how much cost WRR will save from RR. Notice that $\frac{1}{n} \leq \frac{\mathbf{H}_2}{\mathbf{H}_1} \leq 1$. In the degenerated case $\mathbf{H}_2/\mathbf{H}_1 = 1$, WRR does not save any cost from RR. This case occurs only when all arms have identical reward expectations, which is rare and not interesting. However, if $\mathbf{H}_2/\mathbf{H}_1 = 1/n$, by setting $\rho$ to the optimal value in Eq. (6), it is possible to save a substantial portion of pulls. In this scenario, the RR algorithm will iteratively pull all the arms until the arm closest to the threshold $i^*$ confidently determined as outlier or normal. However, the WRR algorithm is able to invest more pulls on arm $i^*$ as it remains active, while pulling other arms for fewer times, only to obtain a more precise estimate of the outlier threshold.

### 4.3 Confidence Interval Instantiation

With different prior knowledge of reward distributions, confidence intervals can be instantiated differently. We introduce the confidence interval for a relatively general scenario, where reward distributions are bounded.

**Bounded distribution.** Suppose the reward distribution of each arm is bounded in $[a, b]$, and $R = b - a$.

According to Hoeffding's inequality and McDiarmid's inequality, we can derive the confidence interval for $y_i$ as

$$\beta_i(m_i, \delta') = R\sqrt{\frac{1}{2m_i}\log\left(\frac{1}{\delta'}\right)}, \quad \beta_\theta(\mathbf{m}, \delta') = R\sqrt{\frac{l(k)}{2h(\mathbf{m})}\log\left(\frac{1}{\delta'}\right)}$$

where $m_i$ is the number of pulls of arm $i$ so far, and $h(\mathbf{m})$ is the harmonic mean of all the $m_i$'s.

**Bernoulli distribution.** In many real applications, each arm returns a binary sample 0 or 1, drawn from a Bernoulli distribution. We use the following confidence intervals heuristically.

We leverage a confidence interval presented in [3], defined as

$$\beta_i(m_i, \delta') = z_{\delta'/2}\sqrt{\frac{\tilde{p}(1 - \tilde{p})}{m_i}}, \quad \beta_\theta(\mathbf{m}, \delta') = \sqrt{\sum_i\left(\frac{k\hat{y}_i}{n\sqrt{\hat{\sigma}_y}} + \frac{1}{n}\right)^2\beta_i^2}$$

where

$$\tilde{p} = \frac{m_i^+ + \frac{z_{\delta'/2}^2}{2}}{m_i + z_{\delta'/2}^2}, \quad z_{\delta'/2} = \text{erf}^{-1}(1 - \delta'/2)$$

$m_i^+$ is the number of samples that equal to 1 among $m_i$ samples, and $z_{\delta'/2}$ is value of the *inverse error function*.

## 5 Experimental Results

In this section, we present experiments to evaluate both the effectivenss and efficiency of proposed algorithms.

### 5.1 Datasets

**Synthetic.** We construct several synthetic datasets with varying number of arms $n = 20, 50, 100, 200$, and varying $k = 2, 2.5, 3$. There are 12 configurations in total. For each configuration, we generate 10 random test cases. For each arm, we draw its reward from a Bernoulli distribution $Bern(y_i)$.

**Twitter.** We consider the following application of detecting outlier locations with respect to keywords from Twitter data. A user has a set of candidate regions $L = \{l_1, \cdots, l_n\}$, and is interested in finding outlier regions where tweets are extremely likely to contain a keyword $w$. In this application, each region corresponds to an arm. A region has an unknown probability of generating a tweet containing the keyword, which can be regarded as a Bernoulli distribution. We collect a Twitter dataset with $1,500,000$ tweets from NYC, associated with its latitude and longitude. We divide the entire space into regions of $2'' \times 2''$ in latitude and longitude respectively. We select $47$ regions with more than $5,000$ tweets as arms and select $20$ keywords as test cases.

## 5.2 Setup

**Methods for comparison.** Since the problem is new, there is no directly comparable solution in existing work. We design two baselines for comparative study.

- *Naive Round-Robin (NRR).* We play arms in a round-robin fashion, and terminate as soon as we find the estimated outlier set $\hat{\Omega}$ has not changed in the last consecutive $1/\delta$ pulls. $\hat{\Omega}$ is defined as in Eq. (3). This baseline reflects how well the problem can be solved by RR with a heuristic termination condition.
- *Iterative Best Arm Identification (IB).* We apply a state-of-the-art best arm identification algorithm [12] iteratively. We first apply it to all $n$ arms until it terminates, and then remove the best arm and apply it to the rest arms. We repeat this process until the current best arm is not in $\hat{\Omega}$, where the threshold function is heuristically estimated based on the current data. We then return the current $\hat{\Omega}$. This is a strong baseline that leverages the existing solution in best-arm identification.

Then we compare them with our proposed two algorithms, Round-Robin (RR) and Weighted Round-Robin (WRR).

**Parameter configurations.** For both of our algorithms, we derived the confidence intervals based on Bernoulli distribution. Since some algorithm takes extremely long time to terminate in certain cases, we place a cap on the total number of pulls. Once an algorithm runs for $10^7$ pulls, the algorithm is forced to terminate and output the current estimated outlier set $\hat{\Omega}$. We set $\delta = 0.1$.

For each test case, we run the experiments for 10 times, and take the average of both the correctness metrics and number of pulls.

## 5.3 Results

**Performance on Synthetic.** Figure 1(a) shows the correctness of each algorithm when $n$ varies. It can be observed that both of our proposed algorithms achieve perfect correctness on all the test sets. In comparison, the NRR baseline has never achieved the desired level of correctness. Based on the performance on correctness, the naive baseline NRR does not qualify an acceptable algorithm, so we only measure the efficiency of the rest algorithms.

We plot the average number of pulls each algorithm takes before termination varying with the number of arms $n$ in Figure 1(b). On all the different configurations of $n$, IB takes a much larger number of pulls than WRR and RR, which makes it 1-3 orders of magnitude as costly as WRR and RR. At the same time, RR is also substantially slower than WRR, with the gap gradually increasing as $n$ increases. This shows our design of additional pulls helps. Figure 1(c) further shows that in 80% of the test cases, WRR can save more than 40% of cost from RR; in about half of the test cases, WRR can save more than 60% of the cost.

**Performance on Twitter.** Figure 2(a) shows the correctness of different algorithms on Twitter data set. As one can see, both of our proposed algorithms qualify the correctness requirement, i.e., the probability of returning the exactly correct outlier set is higher than $1 - \delta$. The NRR baseline is far from reaching that bar. The IB baseline barely meets the bar, and the precision, recall and F1 measures show that its returned result is averagely a good approximate to the correct result, with an average F1 metric close to $0.95$. This once again confirms that IB is a strong baseline.

We compare the efficiency of IB, RR and WRR algorithms in Figure 2(b). In this figure, we plot the cost reduction percentage for both RR and WRR in comparison with IB. WRR is a clear winner. In almost 80% of the test cases, it saves more than 50% of IB's cost, and in about 40% of the test

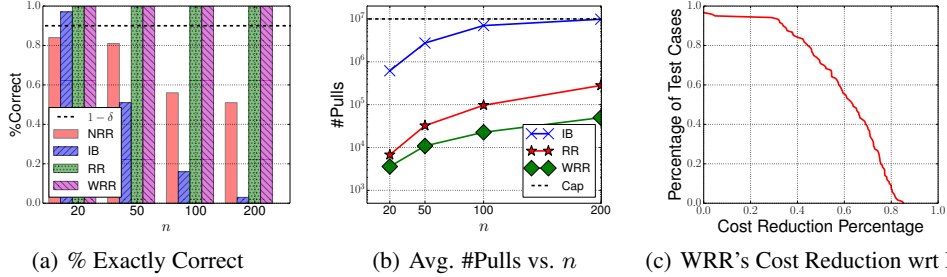

(a) % Exactly Correct       (b) Avg. #Pulls vs. $n$       (c) WRR's Cost Reduction wrt RR

Figure 1: Effectiveness and efficiency studies on Synthetic data set. Cap indicates the maximum number of pulls we allow an algorithm to run.

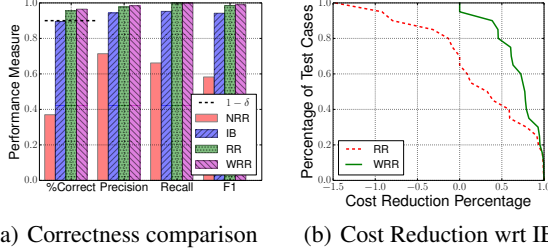

(a) Correctness comparison       (b) Cost Reduction wrt IB

Figure 2: Effectiveness and efficiency studies on Twitter dataset.

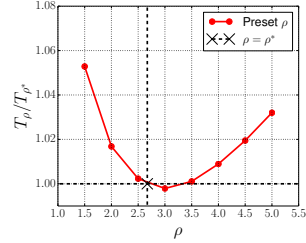

Figure 3: Ratio between avg. #pulls with a given $\rho$ and with $\rho = \rho^*$.

cases, it saves more than 75% of IB's cost. In contrast, RR's performance is comparable to IB. In approximately 30% of the test cases, RR is actually slower than IB and has negative cost reduction, though in another 40% of the test cases, RR saves more than 50% of IB's cost.

**Tuning $\rho$.** In order to experimentally justify our selection of $\rho$ value, we test the performance of WRR on a specific setting of synthetic data set ($n = 15$, $k = 2.5$) with varying preset $\rho$ values. Figure 3 shows the average number of pulls of 10 test cases for each $\rho$ in $\{1.5, 2, \ldots, 5\}$, comparing to the performance with $\rho = \rho^*$ according to Eq. (6). It can be observed that the performance of $\rho = \rho^*$ is very close to the best performance when $\rho = 3$. A further investigation reveals that the $\frac{\mathbf{H_1}}{\mathbf{H_2}}$ of these test cases vary from 3 to 14. Although we choose $\rho^*$ based on an extreme assumption $\frac{\mathbf{H_1}}{\mathbf{H_2}} = n$, its average performance is found to be close to the optimal even when the data do not satisfy the assumption.

## 6 Conclusion

In this paper, we study a novel problem of identifying the outlier arms with extremely high/low reward expectations compared to other arms in a multi-armed bandit. We propose a Round-Robin algorithm and a Weighted Round-Robin algorithm with correctness guarantee. We also upper bound both algorithms when the reward distributions are bounded. We conduct experiments on both synthetic and real data to verify our algorithms. There could be further extensions of this work, including deriving a lower bound of this problem, or extending the problem to a PAC setting.

## Footnotes

*The authors would like to thank anonymous reviewers for their helpful comments.

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
