[Supplementary Material]

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

# Appendices

## A  Correctness (Theorem 1)

We start by confining our discussion into an event, where $y_i$ and $\theta$ do not fall out of the given confidence intervals.

**Lemma 1.** *Suppose we are given confidence interval functions $\beta_i(m_i, \delta')$ for $\forall i$ and $\beta_\theta(\mathbf{m}, \delta')$ related to the number of pulls and an arbitrary error probability $0 < \delta' < 1$. They satisfy*

$$\mathbb{P}(\hat{y}_i - y_i > \beta_i(m_i, \delta')) < \delta', \quad \mathbb{P}(\hat{y}_i - y_i < -\beta_i(m_i, \delta')) < \delta'$$

$$\mathbb{P}(\hat{\theta} - \theta > \beta_\theta(\mathbf{m}, \delta')) < \delta', \quad \mathbb{P}(\hat{\theta} - \theta < -\beta_\theta(\mathbf{m}, \delta')) < \delta'$$

*Define the random event*

$$\mathcal{E} = \Big\{ \hat{y}_i - \beta_i(T, m_i) \leq y_i \leq \hat{y}_i + \beta_i(T, m_i)$$

$$\bigwedge \hat{\theta} - \beta_\theta(T, \mathbf{m}) \leq \theta \leq \hat{\theta} + \beta_\theta(T, \mathbf{m}), \forall i, \forall T \Big\}$$

*Suppose a sequence $S = [I_1, I_2, \ldots]$ is an infinite sequence where $1 \leq I_t \leq n$ is a integer, representing the arm pulled at iteration $t$. If we properly set the tolerance of confidence intervals $\delta'$ according to the current number of iterations, namely*

$$\delta'(T) = \frac{6\delta}{\pi^2(n+1)T^2}$$

*then for any S, we have*

$$\mathbb{P}(\mathcal{E}|S) \geq 1 - \delta$$

*Proof.*

$$1 - \mathbb{P}(\mathcal{E}|S) \leq \sum_{T=1}^{\infty}\left[ \frac{6\delta}{\pi^2(n+1)T^2} + \sum_{i=1}^{n} \frac{6\delta}{\pi^2(n+1)T^2} \right]$$

$$= \frac{6}{\pi^2} \sum_{T=1}^{\infty} \frac{\delta}{T^2} = \delta$$

$\square$

Based on this, it is straightforward to prove Theorem 1.

## B  Confidence Intervals of Bounded Reward Distributions

In this appendix, we show an instantiation of confidence interval when all the reward distributions are bounded. Without loss of generality, suppose they are bounded in $[a, b]$ and $R = b - a$.

We start by a direct application of Hoeffding's inequality to depict the concentration of each arm's expectation estimate.

**Lemma 2** (Hoeffding). *The probability that the difference between $\hat{y}_i$ and $y_i$ is larger than a given constant $t$ can be bounded as:*

$$\mathbb{P}(\hat{y}_i - y_i \geq t) \leq \exp\left( \frac{-2m_i t^2}{R^2} \right) \tag{8}$$

$$\mathbb{P}(\hat{y}_i - y_i \leq -t) \leq \exp\left( \frac{-2m_i t^2}{R^2} \right) \tag{9}$$

We then apply McDiarmid's inequality to describe the concentration of the threshold function estimator.

**Lemma 3.** *The probability that the difference between the estimated threshold function and the expected estimation of the value of the threshold function is larger than a given constant $t$ can be bounded as:*

$$\mathbb{P}(\hat{\theta} - \mathbb{E}\hat{\theta} \geq t) \leq \exp\left(\frac{-2h(\mathbf{m})t^2}{R^2 g(k)}\right) \tag{10}$$

$$\mathbb{P}(\hat{\theta} - \mathbb{E}\hat{\theta} \leq -t) \leq \exp\left(\frac{-2h(\mathbf{m})t^2}{R^2 g(k)}\right) \tag{11}$$

*where $g(k) = (1 + k\sqrt{n-1})^2/n$.*

*Proof.* Consider $\hat{\mu}_y$ to be a function of all the samples $\hat{\mu}_y(\mathbf{x}_1, \cdots, \mathbf{x}_n)$. Since

$$\hat{\mu}_y(\mathbf{x}_1, \cdots, \mathbf{x}_n) = \frac{1}{n}\sum_i\sum_j \frac{x_i^{(j)}}{m_i} \tag{12}$$

Hence

$$\sup_{x_i^{(j)}} \hat{\mu}_y(\mathbf{x}_1, \cdots, \mathbf{x}_n) - \inf_{x_i^{(j)}} \hat{\mu}_y(\mathbf{x}_1, \cdots, \mathbf{x}_n) \leq \frac{b-a}{nm_i} = \frac{R}{nm_i} \tag{13}$$

Consider $\hat{\sigma}_y$ to be a function of all the samples $\hat{\sigma}_y(\mathbf{x}_1, \cdots, \mathbf{x}_n)$. For any given pair of $i$ and $j$, let

$$A_i = \frac{1}{n-1}\sum_{i' \neq i} \hat{y}_{i'}^2$$

$$B_i = \frac{1}{n-1}\sum_{i' \neq i} \hat{y}_{i'}$$

$$M_{ij} = \frac{1}{m_i - 1}\sum_{j' \neq j} x_i^{(j)}$$

where $B_i$ is the mean of all the estimated $\hat{y}_{i'}$ other than $\hat{y}_{i'}$; $(A_i - B_i)^2$ is their standard deviation and therefore is no less than 0; $M_{ij}$ is the mean of all but the $j$-th samples from the $i$-th arm.

We represent the estimated standard deviation $\hat{\sigma}_y$ as a function of random variable $x_i^{(j)}$ and the variables above:

$$\hat{\sigma}_y(\mathbf{x}_1, \cdots, \mathbf{x}_n) = \sqrt{\frac{\sum_i \hat{y}_i^2}{n} - \left(\frac{\sum_i \hat{y}_i}{n}\right)^2}$$

$$= \sqrt{\frac{A_i(n-1) + \hat{y}_i^2}{n} - \left(\frac{B_i(n-1) + \hat{y}_i}{n}\right)^2}$$

$$= \sqrt{\left(1 - \frac{1}{n}\right)\left(\frac{1}{n}(\hat{y}_i - B_i)^2 + (A_i - B_i^2)\right)}$$

$$= \sqrt{1 - \frac{1}{n}}\sqrt{\frac{1}{n}\left(\frac{(m_i - 1)M_{ij} + x_i^{(j)}}{m_i} - B_i\right)^2 + (A_i - B_i^2)}$$

For any given $\mathbf{x}$, we want to bound the maximum possible difference of this function by keeping all variables fixed but only adjusting $x_i^{(j)}$

Let $x_1$, $x_2$ be the value of $x_i^j$ when $\hat{\sigma}_y$ has the maximum and minimum value respectively. For any given $\mathbf{x}_1 \cdots, \mathbf{x}_n$ except $x_i^j$, we have,

$$
\begin{aligned}
&\sup_{x_i^{(j)}} \hat{\sigma}_y(\mathbf{x}_1, \cdots, \mathbf{x}_n) - \inf_{x_i^{(j)}} \hat{\sigma}_y(\mathbf{x}_1, \cdots, \mathbf{x}_n)\\
&\leq \sqrt{1 - \frac{1}{n}} \left[ \sqrt{\frac{1}{n}\left(\frac{(m_i-1)M_{ij}+x_1}{m_i} - B_i\right)^2} - \sqrt{\frac{1}{n}\left(\frac{(m_i-1)M_{ij}+x_2}{m_i} - B_i\right)^2} \right]\\
&\leq \sqrt{\frac{1}{n}\left(1 - \frac{1}{n}\right)} \left[\left|\frac{(m_i-1)M_{ij}+x_1}{m_i} - B_i\right| - \left|\frac{(m_i-1)M_{ij}+x_2}{m_i} - B_i\right|\right]\\
&\leq \sqrt{\frac{1}{n}\left(1 - \frac{1}{n}\right)} \left(\frac{x_1 - x_2}{m_i}\right)\\
&\leq \frac{b-a}{nm_i}\sqrt{n-1} = \frac{R}{nm_i}\sqrt{n-1}
\end{aligned}
$$

Since the threshold function is a linear combination of $\hat{\mu}_y$ and $\hat{\sigma}_y$, it also has bounded difference:

$$
\begin{aligned}
&\sup_{x_i^{(j)}} \hat{\theta}(\mathbf{x}_1, \cdots, \mathbf{x}_n) - \inf_{x_i^{(j)}} \hat{\theta}(\mathbf{x}_1, \cdots, \mathbf{x}_n)\\
&\leq \sup_{x_i^{(j)}} \hat{\mu}(\mathbf{x}_1, \cdots, \mathbf{x}_n) - \inf_{x_i^{(j)}} \hat{\mu}(\mathbf{x}_1, \cdots, \mathbf{x}_n) + \sup_{x_i^{(j)}} k\hat{\sigma}(\mathbf{x}_1, \cdots, \mathbf{x}_n) - \inf_{x_i^{(j)}} k\hat{\sigma}(\mathbf{x}_1, \cdots, \mathbf{x}_n)\\
&\leq \frac{R}{nm_i}\left(1 + k\sqrt{n-1}\right)
\end{aligned}
\tag{14}
$$

Let $c_{ij} = \frac{R}{nm_i}\left(1 + k\sqrt{n-1}\right)$. We can have

$$
\begin{aligned}
\sum_i \sum_j c_{ij}^2 &= \sum_i m_i \frac{R^2}{n^2 m_i^2}\left(1 + k\sqrt{n-1}\right)^2\\
&= \left(1 + k\sqrt{n-1}\right)^2 \frac{R^2}{n^2} \sum_i \frac{1}{m_i}
\end{aligned}
$$

According to McDiarmid's inequality, we can have the following concentration guarantee:

$$
\begin{aligned}
\mathbb{P}(\hat{\theta} - \mathbb{E}\hat{\theta} \geq t) &\leq \exp\left(\frac{-2t^2}{\sum_i \sum_j c_{ij}^2}\right) = \exp\left(\frac{-2n^2 t^2}{R^2\left(1 + k\sqrt{n-1}\right)^2 \sum_i \frac{1}{m_i}}\right)\\
&= \exp\left(\frac{-2h(\mathbf{m})t^2}{R^2 g(k)}\right)
\end{aligned}
\tag{15}
$$

where $g(k) = (1 + k\sqrt{n-1})^2/n$.

Similarly, we have

$$
\mathbb{P}(\hat{\theta} - \mathbb{E}\hat{\theta} \leq -t) \leq \exp\left(\frac{-2h(\mathbf{m})t^2}{R^2 g(k)}\right)
\tag{16}
$$

$\square$

And since $\mathbb{E}\hat{\theta}$ is a biased estimator, we also need to bound its bias $|\mathbb{E}\hat{\theta} - \theta|$.

**Lemma 4.** *The bias of the threshold function estimator can be bounded as:*

$$
\left|\mathbb{E}\hat{\theta} - \theta\right| \leq kR\sqrt{\frac{n-1}{4n}\frac{1}{h(\mathbf{m})}}
\tag{17}
$$

*Proof.* Notice that

$$\mathbb{E}\hat{\theta} - \theta = \mathbb{E}\hat{\mu}_y - \mu_y + k\mathbb{E}\hat{\sigma}_y - k\sigma_y = k\Big(\mathbb{E}\hat{\sigma}_y - \sigma_y\Big)$$

We only need to upper bound $\left|\mathbb{E}\hat{\sigma}_y - \sigma_y\right|$. Let $\hat{e}_i = \hat{y}_i - y_i$, and $\hat{\mu}_e = \hat{\mu}_y - \mu_y$, we have:

$$\mathbb{E}\hat{\sigma}_y = \mathbb{E}\sqrt{\frac{1}{n}\sum_i \Big(y_i - \mu_y + \hat{e}_i - \hat{\mu}_e\Big)^2}$$

$$= \mathbb{E}\sqrt{\frac{1}{n}\left[\sum_i (y_i - \mu_y)^2 + \sum_i 2(y_i - \mu_y)(\hat{e}_i - \hat{\mu}_e) + \sum_i (\hat{e}_i - \hat{\mu}_e)^2\right]}$$

According to the Cauchy–Schwarz inequality,

$$\left|\sum_i (y_i - \mu_y)(\hat{e}_i - \hat{\mu}_e)\right| \leq \sqrt{\sum_i (y_i - \mu_y)^2} \cdot \sqrt{\sum_i (\hat{e}_i - \hat{\mu}_e)^2}$$

Therefore, we can obtain the upper bound for $\mathbb{E}\hat{\sigma}_y$ as

$$\mathbb{E}\hat{\sigma}_y \leq \mathbb{E}\sqrt{\frac{1}{n}\left[\sum_i (y_i - \mu_y)^2 + 2\sqrt{\sum_i (y_i - \mu_y)^2} \cdot \sqrt{\sum_i (\hat{e}_i - \hat{\mu}_e)^2} + \sum_i (\hat{e}_i - \hat{\mu}_e)^2\right]}$$

$$= \mathbb{E}\sqrt{\frac{1}{n}\left(\sqrt{\sum_i (y_i - \mu_y)^2} + \sqrt{\sum_i (\hat{e}_i - \hat{\mu}_e)^2}\right)^2}$$

$$= \sqrt{\frac{1}{n}\sum_i (y_i - \mu_y)^2} + \mathbb{E}\sqrt{\frac{1}{n}\sum_i (\hat{e}_i - \hat{\mu}_e)^2}$$

$$= \sigma_y + \mathbb{E}\sqrt{\frac{1}{n}\sum_i (\hat{e}_i - \hat{\mu}_e)^2}$$

Symmetrically, we can obtain the lower bound for $\mathbb{E}\hat{\sigma}_y$ as

$$\mathbb{E}\hat{\sigma}_y \geq \sigma_y - \mathbb{E}\sqrt{\frac{1}{n}\sum_i (\hat{e}_i - \hat{\mu}_e)^2}$$

Therefore, we only need to bound the term $\mathbb{E}\sqrt{\frac{1}{n}\sum_i (\hat{e}_i - \hat{\mu}_e)^2}$. By applying Jensen's inequality:

$$\mathbb{E}\sqrt{\frac{1}{n}\sum_i (\hat{e}_i - \hat{\mu}_e)^2} \leq \sqrt{\frac{1}{n}\sum_i \mathbb{E}(\hat{e}_i - \hat{\mu}_e)^2}$$

$$= \sqrt{\frac{1}{n}\sum_i \mathbb{E}(\hat{y}_i - y_i)^2 - \mathbb{E}\Big(\sum_i \frac{\hat{y}_i - y_i}{n}\Big)^2}$$

Assuming the variance of the $i$-th arm's distribution is $\sigma_i^2$, when there are $m_i$ samples collected to estimate $\hat{y}_i$, we have $\mathbb{E}[\hat{y}_i^2] = y_i^2 + \frac{\sigma_i^2}{m_i}$. Therefore, the equation becomes

$$\sqrt{\frac{1}{n}\sum_i \mathbb{E}(\hat{e}_i - \hat{\mu}_e)^2} = \sqrt{\frac{n-1}{n^2}\sum_i \frac{\sigma_i^2}{m_i}}$$

Further, since we have assume all the distributions are bounded, we can apply Popoviciu's inequality on variances, which states $\sigma_i^2 \leq R^2/4$. Hence,

$$\sqrt{\frac{n-1}{n^2}\sum_i \frac{\sigma_i^2}{m_i}} \leq R\sqrt{\frac{n-1}{4n}\frac{1}{h(\mathbf{m})}}$$

Combining the above equations, we have

$$\sigma_y - R\sqrt{\frac{n-1}{4n}\frac{1}{h(\mathbf{m})}} \leq \mathbb{E}\hat{\sigma}_y \leq \sigma_y + R\sqrt{\frac{n-1}{4n}\frac{1}{h(\mathbf{m})}} \tag{18}$$

$\square$

From Lemma 3, we can derive with probability $1 - \delta'$,

$$\mathbb{E}\hat{\theta} \leq \hat{\theta} + R\sqrt{\frac{l(k)}{2h(\mathbf{m})}\log\left(\frac{1}{\delta'}\right)}$$

From Lemma 4, we have

$$\theta \leq \mathbb{E}\hat{\theta} + kR\sqrt{\frac{n-1}{4n}\frac{1}{h(\mathbf{m})}}$$

Combining these two equations, we have (with probability $1 - \delta'$)

$$\theta \leq \hat{\theta} + R\sqrt{\frac{g(k)}{2h(\mathbf{m})}\log\left(\frac{1}{\delta'}\right)} + kR\sqrt{\frac{n-1}{4n}\frac{1}{h(\mathbf{m})}}$$

$$\leq \hat{\theta} + R\sqrt{\frac{1}{2h(\mathbf{m})}\log\left(\frac{1}{\delta'}\right)}\left(\sqrt{g(k)} + \sqrt{\frac{k^2}{2\log\left(\frac{1}{\delta'}\right)}}\right)$$

In our algorithm, the error $\delta'$ is set according to a function $\delta'(T)$ to guarantee the validity of the confidence interval through the entire sampling process. According to Eq. (1), we substitute its definition into the inequality above:

$$\theta \leq \hat{\theta} + R\sqrt{\frac{1}{2h(\mathbf{m})}\log\left(\frac{1}{\delta'(T)}\right)}\left(\sqrt{g(k)} + \sqrt{\frac{k^2}{2\log\left(\frac{\pi^2(n+1)T^2}{6\delta}\right)}}\right)$$

Notice that we pull each arm at least once as initialization, so $T \geq n$. Therefore, we can further obtain:

$$\theta \leq \hat{\theta} + R\sqrt{\frac{1}{2h(\mathbf{m})}\log\left(\frac{1}{\delta'(T)}\right)}\left(\sqrt{g(k)} + \sqrt{\frac{k^2}{2\log\left(\frac{\pi^2 n^3}{6\delta}\right)}}\right)$$

$$\leq \hat{\theta} + R\sqrt{\frac{l(k)}{2h(\mathbf{m})}\log\left(\frac{1}{\delta'(T)}\right)}$$

Symmetrically, we can also obtain

$$\theta \geq \hat{\theta} - R\sqrt{\frac{l(k)}{2h(\mathbf{m})}\log\left(\frac{1}{\delta'(T)}\right)}$$

with probability $1 - \delta'$.

## C  Upper Bound of Round-Robin Algorithm (Theorem 2)

We start by a lemma indicating how small the confidence intervals should be could we guarantee a certain arm is inactive.

**Lemma 5.** *At the $T$-th iteration, if $\mathcal{E}$ happens, and arm $i$ is active, then*

$$\beta_i + \beta_\theta \geq \frac{1}{2}\Delta_i \tag{19}$$

*where $\Delta_i = |y_i - \theta|$.*

*Proof.* If arm $i$ is active, then according to the algorithm, it satisfies the condition that

$$\begin{cases} \hat{y}_i - \beta_i < \hat{\theta} + \beta_\theta, & \text{if } \hat{y}_i > \hat{\theta}; \\ \hat{y}_i + \beta_i > \hat{\theta} - \beta_\theta, & \text{otherwise.} \end{cases} \tag{20}$$

Furthermore, if event $\mathcal{E}$ happens, we should have

$$\hat{y}_i - \beta_i \leq y_i \leq \hat{y}_i + \beta_i$$
$$\hat{\theta} - \beta_\theta \leq \theta \leq \hat{\theta} + \beta_\theta$$

If $\hat{y}_i > \hat{\theta}$, then we have

$$\begin{aligned} y_i - \theta &\leq \hat{y}_i + \beta_i - (\hat{\theta} - \beta_\theta) \\ &= \hat{y}_i - \hat{\theta} + \beta_i + \beta_\theta \\ &\leq 2\beta_i + 2\beta_\theta \end{aligned} \tag{21}$$

Hence,

$$\beta_i + \beta_\theta \geq \frac{1}{2}\Delta_i \tag{22}$$

Symmetrically, for $\hat{y}_i \leq \hat{\theta}$ we can also obtain the same result. $\square$

Now we can give a proof of Theorem 2.

*Proof.* In Round-Robin algorithm, at any iteration, we have

$$|m_i - m_j| \leq 1, \forall 1 \leq i, j \leq n \tag{23}$$

where $i$ and $j$ are integers. According to Lemma 5, if $\mathcal{E}$ happens and arm $i$ is still active, we have

$$\beta_i + \beta_\theta \geq \frac{1}{2}\Delta_i \tag{24}$$

Substituting the confidence intervals by their definitions,

$$\frac{1}{2}\Delta_i \leq R\sqrt{\frac{1}{2m_i}\log\left(\frac{1}{\delta'(T)}\right)} + R\sqrt{\frac{l(k)}{2h(\mathbf{m})}\log\left(\frac{1}{\delta'(T)}\right)}$$

$$\Delta_i \leq R\sqrt{2\log\left(\frac{1}{\delta'(T)}\right)}\left(\sqrt{\frac{1}{m_i}} + \sqrt{\frac{l(k)}{h(\mathbf{m})}}\right)$$

$$\leq R\sqrt{2\log\left(\frac{1}{\delta'(T)}\right)}\left(\sqrt{\frac{1}{m^*}} + \sqrt{\frac{l(k)}{m^*}}\right)$$

$$\leq R\sqrt{2\log\left(\frac{1}{\delta'(T)}\right)}\left(\sqrt{\frac{1}{m^*}} + \sqrt{\frac{l(k)}{m^*}}\right) \tag{25}$$

where $m^* = \min_{i'} m_{i'}$. By organizing the above inequality, we can obtain

$$m^* \leq \frac{2R^2}{\Delta_i^2}\log\left(\frac{1}{\delta'(T)}\right)\left(1 + \sqrt{l(k)}\right)^2 \tag{26}$$

Hence, if the algorithm is not yet terminated, then there must be at least an arm $i$ where the condition above holds.

And since in Round-Robin algorithm, for any $i$ at any time we have $m_i \leq m^* + 1$, we can obtain

$$T = \sum_i m_i \leq n(m^* + 1) \tag{27}$$

We analyze the situation when the algorithm is about to terminate. Right before the algorithm's last pull, denote the minimum number of pulls of a certain arm as $\tilde{m}^*$. Notice that the algorithm is not yet terminated, so we still have

$$\tilde{m}^* \leq \frac{2R^2}{\Delta_i^2} \log\left(\frac{1}{\delta'(T)}\right)\left(1 + \sqrt{l(k)}\right)^2$$

$$\leq \frac{2R^2}{\Delta_{i^*}^2} \log\left(\frac{1}{\delta'(T)}\right)\left(1 + \sqrt{l(k)}\right)^2$$

where $\Delta_{i^*} = \min_{i'} \Delta_{i'}$.

After the last pull, the minimum number of pulls of a certain arm $m^* \leq \tilde{m}^* + 1$. So we have

$$T \leq n(m^* + 1) \leq n(\tilde{m}^* + 2)$$

$$\leq \frac{2nR^2}{\Delta_{i^*}^2} \log\left(\frac{\pi^2(n+1)T^2}{6\delta}\right)\left(1 + \sqrt{l(k)}\right)^2 + 2n$$

$$= 2\mathbf{H}_1 R^2 \log\left(\frac{\pi^2(n+1)T^2}{6\delta}\right)\left(1 + \sqrt{l(k)}\right)^2 + 2n$$

According to Lemma 8 in [4], we have

$$T \leq 8R^2 \mathbf{H}_{\mathrm{RR}}\left[\log\left(\frac{2R^2\pi^2(n+1)\mathbf{H}_{\mathrm{RR}}}{3\delta}\right) + 1\right] + 4n \tag{28}$$

$\square$

## D  Upper Bound of Weighted Round-Robin Algorithm (Theorem 3)

**Lemma 6.** *Suppose after $T$ iterations of the algorithm WRR, the $(T+1)$-th iteration will be a regular pull. If random event $\mathcal{E}$ happens, and arm $i$ has no less than $T_i'$ additional pulls, then arm $i$ is not in active set $A$ ($i \notin A$). $T_i'$ is defined as:*

$$T_i' = \frac{\rho - 1}{\rho}\frac{2R^2}{\Delta_i^2}\log\left(\frac{1}{\delta'(T)}\right)\left(1 + \sqrt{l(k)\rho}\right)^2 \tag{29}$$

*where $\Delta_i = |y_i - \mathbb{E}\hat{\theta}|$.*

*Proof.* According to Lemma 5, if arm $i$ is active and $\mathcal{E}$ happens, we have $\beta_i + \beta_\theta \geq \Delta_i/2$. By substituting the confidence intervals by their definitions, we have:

$$\frac{1}{2}\Delta_i \leq R\sqrt{\frac{1}{2m_i}\log\left(\frac{1}{\delta'(T)}\right)} + R\sqrt{\frac{l(k)}{2h(\mathbf{m})}\log\left(\frac{1}{\delta'(T)}\right)}$$

$$\Delta_i \leq R\sqrt{2\log\left(\frac{1}{\delta'(T)}\right)}\left(\sqrt{\frac{1}{m_i}} + \sqrt{\frac{l(k)}{h(\mathbf{m})}}\right)$$

$$\leq R\sqrt{2\log\left(\frac{1}{\delta'(T)}\right)}\left(\sqrt{\frac{1}{m_i}} + \sqrt{\frac{l(k)}{m^*}}\right) \tag{30}$$

where $m^* = \min_{i'} m_{i'}$. Among the $m_i$ pulls for arm $i$, suppose $m_{i,r}$ are regular pulls, and $m_{i,a}$ are additional pulls, $m_{i,r} + m_{i,a} = m_i$. According to the algorithm, when the flag of additional pull is cleared, and if arm $i$ is still in the active set, then there must be

$$\rho(m_{i,r} - 1) \leq m_i = m_{i,r} + m_{i,a} < \rho(m_{i,r} - 1) + 1 \tag{31}$$

And since the regular pulls follow a round-robin strategy, we should have $m^* \geq m_{i,r} - 1$. Hence,

$$\Delta_i \leq R\sqrt{2\log\left(\frac{1}{\delta'(T)}\right)}\left(\sqrt{\frac{1}{m_i}} + \sqrt{\frac{l(k)}{m_{i,r} - 1}}\right)$$

$$\leq R\sqrt{2\log\left(\frac{1}{\delta'(T)}\right)}\left(\sqrt{\frac{1}{m_i}} + \sqrt{\frac{l(k)\rho}{m_i}}\right)$$

$$m_i \leq \frac{2R^2}{\Delta_i^2}\log\left(\frac{1}{\delta'(T)}\right)\left(1 + \sqrt{l(k)\rho}\right)^2$$

From Eq. (31) we can have

$$m_{i,a} < \frac{\rho - 1}{\rho}m_i \tag{32}$$

which leads to

$$m_{i,a} < \frac{\rho - 1}{\rho}\frac{2R^2}{\Delta_i^2}\log\left(\frac{1}{\delta'(T)}\right)\left(1 + \sqrt{l(k)\rho}\right)^2 = T_i' \tag{33}$$

This is a contradiction to the condition $m_{i,a} \geq T_i'$. Therefore arm $i$ should not be pulled additionally in the next iteration. $\qquad\square$

Next we bound the number of regular pulls for each arm. For the convenience of analysis, we analyze the situation when the algorithm just finishes a "total iteration", when the next $(T + 1)$-th iteration will be a regular pull for arm 1. In this case the number regular pulls for all arms will be the same, denoted as $m_r$. Let $i^* = \arg\min_i \Delta_i$, we have the following theorem.

**Lemma 7.** *Suppose after $T$ iterations the $(T + 1)$-th iteration will be a regular pull for arm 1. If the algorithm is not yet terminated and event $\mathcal{E}$ happens, then the number of regular pulls for any arms should be no more than $T_r'$, where*

$$T_r' = \frac{1}{\rho}\frac{2R^2}{\Delta_{i^*}^2}\log\left(\frac{1}{\delta'(T)}\right)\left(1 + \sqrt{l(k)\rho}\right)^2 + 1 \tag{34}$$

*Proof.* Since the algorithm is not yet terminated, then the active set of arms $A$ is non-empty. Suppose arm $i \in A$, then

$$m_i \leq \frac{2R^2}{\Delta_i^2}\log\left(\frac{1}{\delta'(T)}\right)\left(1 + \sqrt{l(k)\rho}\right)^2$$

And since the $(T + 1)$-th iteration will be a regular pull, the inequality of Eq. (31) stands. Hence,

$$m_r = m_{i,r} \leq \frac{m_i}{\rho} + 1$$

$$\leq \frac{1}{\rho}\frac{2R^2}{\Delta_i^2}\log\left(\frac{1}{\delta'(T)}\right)\left(1 + \sqrt{l(k)\rho}\right)^2 + 1$$

$$\leq \frac{1}{\rho}\frac{2R^2}{\Delta_{i^*}^2}\log\left(\frac{1}{\delta'(T)}\right)\left(1 + \sqrt{l(k)\rho}\right)^2 + 1$$

$\qquad\square$

Now we prove Theorem 3

*Proof.* According to Theorem 1, as long as event $\mathcal{E}$ happens, the returned results will be correct. And since for any possible pulling sequences, the probability of event $\mathcal{E}$ is no less than $(1 - \delta)$, thus the returned result will be correct with probability $(1 - \delta)$.

As for bounding the total number of pulls $T$. Notice that

$$T = \sum_i m_i = \sum_i m_{i,a} + \sum_i m_{i,r} \tag{35}$$

we bound the number of additional and regular pulls respectively.

We start by bounding the final number of additional pulls for each arm. For any arm $i$, when the algorithm terminates, denote the iteration of its last regular pull when it is in the active set as $T_{i,a}$. Then we can apply Lemma 6 to the $(T_{i,a} - 1)$-th iteration, as well as the fact that the following last several additional pulls on arm $i$ will not exceed $\rho$, thus

$$m_{i,a} \leq \frac{\rho - 1}{\rho} \frac{2R^2}{\Delta_i^2} \log\left(\frac{1}{\delta'(T_{i,a} - 1)}\right)\left(1 + \sqrt{l(k)\rho}\right)^2 + \rho$$

$$\leq \frac{\rho - 1}{\rho} \frac{2R^2}{\Delta_i^2} \log\left(\frac{1}{\delta'(T)}\right)\left(1 + \sqrt{l(k)\rho}\right)^2 + \rho$$

We then bound the final number of regular pulls for each arm. Since the regular pulls follow the round-robin strategy, we denote the iteration of the last regular pull on arm 1 as $T_{1,r}$. Then we apply Lemma 7 to the $(T_{1,r} - 1)$-th iteration, and using the fact that there will be no more than 1 more regular pulls for each arm before the algorithm terminates. Thereby we obtain

$$m_{r,a} \leq \frac{1}{\rho} \frac{2R^2}{\Delta_{i^*}^2} \log\left(\frac{1}{\delta'(T_{1,r} - 1)}\right)\left(1 + \sqrt{l(k)\rho}\right)^2 + 2$$

$$\leq \frac{1}{\rho} \frac{2R^2}{\Delta_{i^*}^2} \log\left(\frac{1}{\delta'(T)}\right)\left(1 + \sqrt{l(k)\rho}\right)^2 + 2$$

Thus, when the algorithm terminates, we should have

$$T = \sum_i m_{i,a} + \sum_i m_{i,r}$$

$$\leq \left(\frac{\mathbf{H}_1}{\rho} + \frac{(\rho - 1)\mathbf{H}_2}{\rho}\right)2R^2 \log\left(\frac{1}{\delta'(T)}\right)\left(1 + \sqrt{l(k)\rho}\right)^2 + (\rho + 2)n$$

Substituting $\delta'(T)$ by its definition, we have

$$T \leq \left(\frac{\mathbf{H}_1}{\rho} + \frac{(\rho - 1)\mathbf{H}_2}{\rho}\right)2R^2 \log\left(\frac{\pi^2(n + 1)T^2}{6\delta}\right)\left(1 + \sqrt{l(k)\rho}\right)^2 + (\rho + 2)n$$

$$= 2R^2\mathbf{H}_{\text{WRR}} \log\left(\frac{\pi^2(n + 1)T^2}{6\delta}\right) + (\rho + 2)n$$

Therefore, according to Lemma 8 in [4], we have

$$T \leq 8R^2\mathbf{H}_{\text{WRR}}\left[\log\left(\frac{2R^2\pi^2(n + 1)\mathbf{H}_{\text{WRR}}}{3\delta}\right) + 1\right] + 2(\rho + 2)n \tag{36}$$

$\square$

## E  Confidence Intervals of Bernoulli Reward Distributions

In many real applications, each arm returns a binary sample 0 or 1, drawn from a Bernoulli distribution. In this appendix, we show an heuristic instantiation of confidence interval when all the reward distributions are Bernoulli distributions. We optimize the confidence interval on this specific case with some approximation.

We leverage a confidence interval presented in [3], defined as

$$\beta_i(m_i, \delta') = z_{\delta'/2}\sqrt{\frac{\tilde{p}(1 - \tilde{p})}{m_i}} \tag{37}$$

where

$$\tilde{p} = \frac{m_i^+ + \frac{z_{\delta'/2}^2}{2}}{m_i + z_{\delta'/2}^2}$$

$m_i^+$ is the number of samples that equal to 1 among $m_i$ samples, and $z_{\delta'/2}$ is value of the *inverse error function*:

$$z_{\delta'/2} = \text{erf}^{-1}(1 - \delta'/2)$$

As for the confidence interval of the outlier threshold, we simply apply the propagation of uncertainty, where

$$\beta_\theta(\mathbf{m}, \delta') = \sqrt{\sum_i \left(\frac{\partial \theta}{\partial y_i}\right)^2 \beta_i^2}$$

$$= \sqrt{\sum_i \left(\frac{k y_i}{n \sqrt{\sigma_y}} + \frac{1}{n}\right)^2 \beta_i^2} \approx \sqrt{\sum_i \left(\frac{k \hat{y}_i}{n \sqrt{\hat{\sigma}_y}} + \frac{1}{n}\right)^2 \beta_i^2} \tag{38}$$

## F   Case Study of Twitter Experiments

(a) Ground-truth $y_i$'s         (b) # pulls by RR         (c) # pulls by WRR

Figure 4:  Case study on Twitter dataset for keyword "stadium." Each plotted square represents a region regarded as an arm in our experiments. Darker color indicates higher value.

We conduct a detailed case study to compare the behavior of RR and WRR. For keyword "stadium", Figure 4(a) presents the parameters $y_i$'s of different regions (arms). The largest value appears at a football stadium, which is MetLife stadium, followed by two baseball fields (Yankee stadium and Citi field) and a historic stadium site (Forest Hill). However, only the region containing MetLife stadium is an outlier by $2\sigma$ rule, probably due to the difference of the size between a football stadium and a baseball field, and the tweeting frequency. Figure 4(b) shows the number of pulls on each arms when the RR algorithm terminates. As expected, all the arms are pulled for almost the same number, which is more than 500. In comparison, as shown in Figure 4(c), when WRR terminates, only the arm containing Yankee stadium is pulled for around 500 times, while all the other arms are only pulled for fewer than 200 times. This is because the WRR algorithm focuses more on the arms closer to the outlier threshold, which are harder to be determined as outlier or normal. It saves a lot of iterations on other arms.