[Reviews · NeurIPS 2017]

Reviewer 1



The paper introduces the problem of finding all the "outlier arms" within a multi-armed bandit framework, where an arm is considered to be an outlier if its mean is at least k\sigma more than the average of the set of means (of the arms), where \sigma is the standard deviation within the set of means and k is some input parameter. The authors also provide an algorithm for the PAC setting, and upper bound its sample complexity. The problem with the paper is that the model seems to be just another variant of top-k arm selection with no real motivation. What is the task where it makes more sense to find the outlier arms instead of finding the best 5 arms, or the ones with mean above some threshold? Additionally, the analysis does not seem to reveal some novel insights either (the estimate of \sigma is straightforward as well); and the bounds are not even attempted to be made tighter than what follows from straightforward application of existing methods. Some minor issues and additional questions: - l66-70: what is the difference between (a) and (c)? - in the definition of \sigma and \hat{\sigma} below lines 116 and 119: there should be a sum over the i index. - the sample complexity bound seems to be proportional to k^2. Is this not suboptimal? - line 197: H_2 and H_1 should be bold face. - line 236: what is a region in this context? - line 252: how do you construct and update the set \hat{Omega} for this algorithm? - Why do you present results of different nature in Figure 1 and in Figure 2? Why would it be meaningful to measure the performance differently for the synthetic and for the twitter data? - How did you obtain Equation (15)? It does not seem to follow from (14). - What are the variables you applied McDiarmid's inequality on to obtain (17)? ******* The current emphasis is on the parts that are more straightforward. Highlighting the technical difficulties that the authors had to overcome could significantly help appreciating this work. Additionally, the motivation should be significantly improved upon.

Reviewer 2



This work is concerned with a variant of the multi-armed bandit problem, and more specifically a pure exploration, fixed confidence problem. Here, the objective is to identify with probability at least 1-\delta the arms that are over a threshold, using as few samples as possible. The twist, compared to a setting introduced in [1], is that this threshold is not known a priori by the learner, and instead depends on the means of the arms (unknown to the learner) and an outlier parameter k (known to the learner). With respect to the existing literature, this game can be seen as a pure exploration bandit problem with a double exploration trade-off: first, pulling arms in a balanced way as to estimate each mean to improve the estimate of the (unknown) threshold \hat \theta and second, pulling arms whose means are close to \hat \theta in order to determine whether they belong to the set of outliers. If \theta were known to the learner, it is well understood that one would require roughly 1/(y_i - \theta)^2 samples to determine to which set arm i belongs. To solve this problem, two algorithms are introduced. The first one is a simple uniform exploration algorithm, that comes down to a test (making use of confidence intervals on the estimated threshold and means) as to discover which arms are outliers with high probability. The second algorithm is a distorted version of this algorithm, which overpulls (with respect to the uniform sampling) arms that are still active (i.e. we have not yet identified in which set they belong). The analysis shows that careful tuning of this distortion parameter \rho leads to improved performances over the uniform sampling algorithm. The tuning of this parameter with respect to the complexity measures H_1 (worst case complexity) and H_2 (which is similar to the usual complexity measure in pure exploration problems) is discussed. Extensive experiments are conducted to validate the theoretical results, on both a synthetic and real dataset. I believe there is a typo in the definition of \sigma_y (there should be a sum). Some references are missing see e.g. [1] and [2,3]. In [2,3] the objective is to estimate as well as possible all the means. On the other hand [1] introduced the thresholding bandit problem. Though the lower-bound construction therein does not apply trivially (it is harder to construct problems for the lower-bound given all the interdependencies between the means in this setting), in spirit the complexity terms seem very related, as one should not expect to have a complexity for this problem lower than H_2 (as defined in this work). It would be interesting to see an even more adaptive strategy to tackle this problem, maybe with a different definition for the (problem dependent) threshold if this one proves too difficult for natural adaptivity. I don’t understand the 99% claim in the abstract, you should tone it down or explain it better in the experiments section. Regarding the experiments, I believe that \delta should be set to a much lower value (0.01) and correspondingly they should be repeated at least 1000 times (even if \delta is not changed, experiments should be averaged over many more than 10 runs). I believe that an interesting baseline could be one where the threshold is given a priori to the learner, and it can simply successively accept or reject arms with confidence parameter \delta/n (to account for the union bound). This strategy should be the hardest baseline (which we don’t expect to beat, but it shows the possible improvement). Some further: they could be repeated with different values of \rho, and the \rho parameter used here should be discussed. It would be interesting to see how the “worst-case” \rho parameter as described in (6) performs against a perfectly tuned \rho, on different problems. The synthetic experiments could be run instead on carefully crafted instances that correspond to different extreme problems. Overall, I believe this new problem is well motivated, and the two algorithms introduced are natural strategies to tackle this problem which should serve as strong baselines for future work. The paper is written in a very clear and clean fashion, and it is easy to follow. One weakness is that the algorithms and techniques used are rather basic as the problem itself is hard to tackle. Moreover, a fixed budget transposition of the setting seems like challenging problem to solve in the future. [1] Locatelli, Andrea, Maurilio Gutzeit, and Alexandra Carpentier. "An optimal algorithm for the thresholding bandit problem." Proceedings of the 33rd International Conference on International Conference on Machine Learning-Volume 48. JMLR. org, 2016. [2] Carpentier, Alexandra, A Lazaric, M Ghavamzadeh, R Munos, Peter Auer "Upper-confidence-bound algorithms for active learning in multi-armed bandits." International Conference on Algorithmic Learning Theory. Springer, Berlin, Heidelberg, 2011. [3] Antos, A., Grover, V., Szepesvári, C.: Active learning in heteroscedastic noise. Theoretical Computer Science 411, 2712–2728 (2010)

Reviewer 3



The paper proposes a new bandit problem: among n arms, the learner must find those whose expected payoff is above a certain problem-dependent threshold that is unknown. Concretely, she must identify the outliers defined as the arms whose empirical average is above \mu + k\sigma where \mu is the overall average of all the means of the arms of the problem and \sigma is the standard deviation. The problem by itself is new and interesting as it cannot be trivially solve by existing best-arm identification algorithms. The paper exposes the idea very clearly and is easy to read overall. My comments are the following : - The definition of \sigma is broken, the sum is missing. - The RR algorithm does not look like a bandit algorithm stricto sensu. There is no real adaptivity in the sampling so it looks ore like sequential statistical tests. Can you comment on the use of the Generalized Likelihood Ratio statistic as in "Optimal Best Arm identification" by (Garivier & Kaufmann, 2016) ? Their case is parametric but as you consider bounded distributions in most of your results, it does not seem more restrictive. - One missing result of this paper is a Lower Bound on the expected sample complexity. One would like to have at least an idea of how it compares to lower bounds on other problems. I think this problem is not trivial at all so it might be a very open question but do you have an idea of how to obtain such a result ? - I am surprised that the chosen 'state-of-the-art' algorithm that is used to compute the IB base line is CLUCB by (Chen et al, 2014). In your case, you only want to identify sequentially the best arm of the remaining ones so it seems to me that a standard (not combinatorial) algorithm would work. Why not use the Track and Stop by (Garivier & Kaufmann, 2016) which has been proven optimal and whose code is available online ? typo: l.31 'If' It is overall a good paper, though I'm curious to read answers to my comments. I could increase my rating after the author's rebuttal. %% After rebuttal %%% I read the rebuttal and I thank the authors for their answers to my concerns and their precise comments to all the reviewers. I maintain my overall good appreciation of the paper which, I believe, presents an interesting new setting.